# ON DISENTANGLED REPRESENTATIONS EXTRACTED FROM PRETRAINED GANS

## ABSTRACT

Constructing disentangled representations is known to be a difficult task, especially in the unsupervised scenario. The dominating paradigm of unsupervised disentanglement is currently to train a generative model that separates different factors of variation in its latent space. This separation is typically enforced by training with specific regularization terms in the model's objective function. These terms, however, introduce additional hyperparameters responsible for the trade-off between disentanglement and generation quality. While tuning these hyperparameters is crucial for proper disentanglement, it is often unclear how to tune them without external supervision.

This paper investigates an alternative route to disentangled representations. Namely, we propose to extract such representations from the state-of-the-art GANs trained without disentangling terms in their objectives. This paradigm of *post hoc* disentanglement employs little or no hyperparameters when learning representations, while achieving results on par with existing state-of-the-art, as shown by comparison in terms of established disentanglement metrics, fairness, and the abstract reasoning task. All our code and models are publicly available[1].

## 1 INTRODUCTION

Unsupervised learning of disentangled representations is currently one of the most important challenges in machine learning. Identifying and separating the factors of variation for the data at hand provides a deeper understanding of its internal structure and can bring new insights into the data generation process. Furthermore, disentangled representations are shown to benefit certain downstream tasks, e.g., fairness (Locatello et al., 2019a) and abstract reasoning (van Steenkiste et al., 2019). Since the seminal papers on disentanglement learning, such as InfoGAN (Chen et al., 2016) and $\beta$-VAE (Higgins et al., 2017), a large number of models were proposed, and this problem continues to attract much research attention (Alemi et al., 2016; Chen et al., 2018; Burgess et al., 2017; Kim & Mnih, 2018; Kumar et al., 2018; Rubenstein et al., 2018; Esmaeili et al., 2019; Mathieu et al., 2019; Rolinek et al., 2019; Nie et al., 2020; Lin et al., 2020).

The existing models achieve disentanglement in their latent spaces via specific regularization terms in their training objectives. Typically, these terms determine the trade-off between disentanglement and generation quality. For example, for $\beta$-VAE (Higgins et al., 2017), one introduces the KL-divergence regularization term that constrains the VAE bottleneck's capacity. This term is weighted by the $\beta$ multiplier that enforces better disentanglement for $\beta > 1$ while resulting in worse reconstruction quality. Similarly, InfoGAN utilized a regularization term approximating the mutual information between the generated image and factor codes. As has been shown in the large scale study Locatello et al. (2019b), hyperparameter values can critically affect the obtained disentanglement. In the unsupervised setting, the values of ground truth latent factors utilized by disentanglement metrics are unknown, and thus selection of correct hyperparameters becomes a nontrivial task.

In this paper, we investigate if disentangled representations can be extracted from the pretrained non-disentangled GAN models, which currently provide the state-of-the-art generation quality (Karras et al., 2020). These GANs are trained without disentanglement terms in their objectives; therefore, we do not need to tune the hyperparameters mentioned above. Our study is partially inspired by a very recent line of works on controllable generation (Voynov & Babenko, 2020; Shen & Zhou, 2020;

---

[1]https://bit.ly/3ipb6dW

Härkönen et al., 2020; Peebles et al., 2020), which explore the latent spaces of pretrained GANs and identify the latent directions useful for image editing. The mentioned methods operate without external supervision, therefore, are valid to use in the unsupervised disentanglement. As shown by the comparison on the common benchmarks, the proposed *post hoc* disentanglement is competitive to the current state-of-the-art in terms of existing metrics, becoming an important alternative to the established "end-to-end" disentanglement.

Overall, our contributions are the following:

- We investigate an alternative paradigm to construct disentangled representations by extracting them from non-disentangled models. In this setting, one does not need to tune hyperparameters for disentanglement regularizers.

- We bridge the fields of unsupervised controllable generation and disentanglement learning by using the developments of the former to benefit the latter. As a separate technical contribution, we propose a new simple technique, which outperforms the existing prior methods of controllable generation.

- We extensively evaluate all the methods on several popular benchmarks employing commonly used metrics. In most of the operating points, the proposed post hoc disentanglement successfully reaches competitive performance.

## 2 RELATED WORK

### 2.1 DISENTANGLED REPRESENTATIONS

Learning disentangled representation is a long-standing goal in representation learning (Bengio et al., 2013) useful for a variety of downstream tasks (LeCun et al., 2004; Higgins et al., 2018; Tschannen et al., 2018; Locatello et al., 2019a; van Steenkiste et al., 2019). While there is no strict definition of disentangled representation, we follow the one considered in (Bengio et al., 2013): disentangled representation is a representation where a change in one dimension corresponds to the change only in one factor of variation while leaving other factors invariant. Natural data is assumed to be generated from independent factors of variations, and well-learned disentangled representations should separate these explanatory sources.

The most popular approaches so far were based on variational autoencoders (VAEs). Usually, to make representations "more disentangled", VAEs objectives are enriched with specific regularizers (Alemi et al., 2016; Higgins et al., 2017; Chen et al., 2018; Burgess et al., 2017; Kim & Mnih, 2018; Kumar et al., 2018; Rubenstein et al., 2018; Esmaeili et al., 2019; Mathieu et al., 2019; Rolinek et al., 2019). The general idea behind these approaches is to enforce an aggregated posterior to be factorized, thus providing disentanglement.

Another line of research on disentangled representations is based on the InfoGAN model (Chen et al., 2016). InfoGAN is an unsupervised model, which adds an extra regularizer to GAN loss to maximize the mutual information between the small subset of latent variables (factor codes) and observations. In practice, the mutual information loss is approximated using an encoder network via Variational Information Maximization. InfoGAN-CR(Lin et al., 2020) is a modification of InfoGAN that employs the so-called *contrastive regularizer (CR)*, which forces the elements of the latent code set to be visually perceptible and distinguishable between each other. A very recently proposed InfoStyleGAN model (Nie et al., 2020) incorporates similar ideas into the state-of-the-art StyleGAN architecture, allowing for producing both disentangled representations and achieving excellent visual quality of samples.

In contrast to these approaches, we use **no regularizers or additional loss functions** and simply study state-of-the-art GANs trained in a conventional manner.

### 2.2 CONTROLLABLE GENERATION

Based on rich empirical evidence, it is believed that the latent space of GANs can encode meaningful semantic transformations, such as orientation, appearance, or presence of objects in scenes, of generated images via *vector arithmetic* (Radford et al., 2016; Zhu et al., 2016; Bau et al., 2019; Chen et al., 2016). This means that for an image produced by some latent code, such a transformation can be obtained by simply shifting this latent code in a certain carefully constructed direction, independent from the chosen latent code. E.g., in the case of human faces, we may have separate directions

for such factors as hair color, age, gender. The main applications of this property have been in the field of *controllable generation*, i.e., building software to allow a user to manipulate an image to achieve a certain goal while keeping the result photorealistic. Powerful generative models are an appealing tool for this task since the generated images lie on the image manifold by construction. To manipulate a *real image*, the standard approach is to invert the generator, i.e., to compute a latent code that approximately corresponds to the desired image, and then apply previously discovered directions (Shen et al., 2020).

The discovery of directions that allow for interesting image manipulations is a nontrivial task, which, however, can be performed in an unsupervised manner surprisingly efficiently (Voynov & Babenko, 2020; Shen & Zhou, 2020; Härkönen et al., 2020; Peebles et al., 2020). In the heart of these methods lies the idea that the deformations produced by these directions should be as *distinguishable* as much as possible, which is achieved via maximizing a certain generator–based loss function or by training a separate regressor network attempting to differentiate between them. We thoroughly discuss these approaches further in the text. An important common feature of these methods is that they do not depend on sensitive hyperparameters or even do not have them at all, which makes them appealing for usage in unsupervised settings.

Contrary to previous applications of such interpretable directions, we attempt to show they allow us to solve a more fundamental task of building disentangled representations, useful in a variety of downstream tasks.

## 3 Two-stage disentanglement using pretrained GANs

In this section, we discuss how disentangled representations of data can be learned with a two-step procedure. Briefly, it can be described as follows. First, we search for a set of $k$ orthogonal interpretable directions in the latent space of the pretrained GAN in an *unsupervised* manner. This step is performed via one of the methods of controllable generation described below. These directions can be considered as the first $k$ vectors of a *new basis in the latent space*. By (virtually) completing it to a full orthogonal set of vectors, we can obtain (presumably, disentangled) representations of *synthetic* points by a simple change of bases and truncating all but the first $k$ coordinates; this can be computed by single matrix multiplication. To obtain such representations for *real data*, we can now train an encoder on a synthetic dataset where targets are constructed using the procedure above. We stick to orthogonal directions for several reasons. Experimentally, it has been shown that this constraint does not significantly affect the quality of discovered directions and is imposed by construction in several further discussed methods. Additionally, it makes the formulas less cumbersome.

**Reminder on style–based architectures.** Traditionally, the generator network in GANs transforms a latent code $\mathbf{z} \in \mathcal{N}(0, 1)$ to an image $\mathbf{x} \in \mathbb{R}^{C \times H \times W}$. Contrary to this, style–based generators introduce a so-called *style network*, usually realized as a trainable MLP, which "preprocesses" random latent codes to the so-called "style vectors" (elements of the *style space*). The obtained style vectors are, in turn, fed into the convolutional layers of the generator. It has been shown (Karras et al., 2019) that direct manipulations over the style vectors, rather than latent codes themselves, leads to visually plausible image interpolations and image mixing results. Intuitively, the style network performs a "rectification" of the latent space so that the new latent codes are more interpretable and disentangled. In this paper, we work with the style–based generators and perform image editing in the style space, denoted by $\mathcal{W}$; its elements are denoted as $\mathbf{w}$. Let us now discuss the steps mentioned above in more detail. Recall, that we are interested in finding directions $\boldsymbol{n}$ such that $G(\boldsymbol{w}')$ with $\boldsymbol{w}' = \boldsymbol{w} + \alpha \boldsymbol{n}$ performs a certain interpretable deformation of the image $G(\boldsymbol{w})$. We now thoroughly describe the existing approaches to obtaining them in an unsupervised manner as well as various hyperparameters one needs to specify for each method.

### 3.1 Discovering interpretable directions

We consider several recently proposed methods: `ClosedForm` (Shen & Zhou, 2020), `GANspace` (Härkönen et al., 2020), `LatentDiscovery` (Voynov & Babenko, 2020). Inspired by these methods, we also propose another family of methods termed `DeepSpectral`.

**ClosedForm (CF).** The authors of the `ClosedForm` method propose to move along the singular vectors of the first layer of generator. More specifically, for the transformation in the first layer given

by the matrix $\boldsymbol{A}$ (e.g., this may be the weight in a fully-connected or a transposed convolutional layer), the direction $\boldsymbol{n}$ is found as

$$\boldsymbol{n}^* = \operatorname*{arg\,max}_{\{\boldsymbol{n} \in \mathbb{R}^D : \boldsymbol{n}^T \boldsymbol{n} = 1\}} \|\boldsymbol{A}\boldsymbol{n}\|_2^2. \tag{1}$$

All local maximas of Equation (1) form the set of singular vectors of the matrix $\boldsymbol{A}$, and the authors propose to choose $k$ singular vectors, associated with the corresponding $k$ highest singular values. For the style–based GANs, the matrix $\boldsymbol{A}$ is obtained by concatenating the style mapping layers of each convolutional block.
**Hyperparameters:** this method is hyperparameter–free and requires only a pretrained GAN model.

**GANspace (GS).** This method searches for important, meaningful directions by performing PCA in the style space of StyleGAN. Specifically, these directions are found in the following manner: first, randomly sampled noise vectors $\boldsymbol{z}_{1:N} \in \mathcal{N}(0, 1)$ are converted into style vectors $\boldsymbol{w}_{1:N}$. The interpretable directions then correspond to principal axes of the set $\boldsymbol{w}_{1:N}$; in practice, we consider top $k$ directions according to the corresponding singular values.
**Hyperparameters:** for this approach, we only need to provide the number of sampled points which can be taken fixed and large, as well as a random seed for sampling.

**LatentDiscovery (LD).** `LatentDiscovery` is an unsupervised model-agnostic procedure for identifying interpretable directions in the GAN latent space. Informally speaking, this method searches for a set of directions that can be easily distinguished from one another. The resulting directions are meant to represent independent factors of generated images and include human-interpretable representations.
The trainable components are the following: a matrix $\boldsymbol{N} \in \mathbb{R}^{D \times k}$ and a reconstructor network $R$, which evaluates the pair $(G(\boldsymbol{w}), G(\boldsymbol{w} + \boldsymbol{N}(\epsilon \boldsymbol{e}_k)))$. The reconstructor model aims to recover the shift in the latent space corresponding to the given image transformation. These two components are optimized jointly by minimizing the following objective function:

$$\boldsymbol{N}^*, R^* = \operatorname*{arg\,min}_{\boldsymbol{N}, R} \mathbb{E}_{\boldsymbol{w}, k, \epsilon} L(\boldsymbol{N}, R) = \operatorname*{arg\,min}_{\boldsymbol{N}, R} \mathbb{E}_{\boldsymbol{w}, k, \epsilon} \left[ L_{cl}(k, \hat{k}) + \lambda L_r(\epsilon, \hat{\epsilon}) \right]. \tag{2}$$

Here, $L_{cl}(\cdot, \cdot)$ is a reconstructor classification loss (cross-entropy function), $L_r(\cdot, \cdot)$ – regression term (mean absolute error), which forces shifts along found directions to be more continuous. This method utilizes a number of hyperparameters; however, as was shown in Voynov & Babenko (2020), it is quite stable, and the default values provide good quality across various models and datasets.
**Hyperparameters:** the hyperparameters include the number of latent directions $k$ and the multiplier of the reconstructor term $\lambda$; additionally, we can select different architectures for the regressor, as well as different training hyperparameters and random seed for initialization.

**DeepSpectral (DS).** We propose a novel approach to finding interpretable directions in the GAN latent space. Our motivation is as follows. While **CF** and **GS** both produce decent results, they effectively ignore all the layers in the generator but the first few ones. We hypothesize that by study-ing outputs of deeper *intermediate* layers of the generator, one can obtain a richer set of directions unavailable for these methods. Concretely, we propose the following simple approach. Let $G^{(i)}(\boldsymbol{w})$ denote the output of the $i$-th hidden layer of the generator. In order to obtain $k$ directions in the latent space, we compute $k$ singular vectors of $J_{G^{(i)}}(\boldsymbol{w})$ with the highest singular values (at some fixed point $\boldsymbol{w}$). Here, $J_{G^{(i)}}$ denotes the *Jacobian matrix* of a mapping. In a way, our approach generalizes **CF** since a linear map and its Jacobian coincide (when bias is zero). By using automatic differenti-ation and an iterative approach to computing Singular Value Decomposition (thus, we do not form the full Jacobian matrix), such directions can be found basically instantly (Khrulkov & Oseledets, 2018). The only hyperparameters in this approach are the choice of the layer and the choice of the base point $\boldsymbol{w}$ to compute the Jacobian. We experimentally verify the benefits of **DS** by considering various intermediate layers in Section 4.
**Hyperparameters:** we need to specify the layer and the base point $\boldsymbol{w}$.

Another recently proposed method (Peebles et al., 2020) searches for interpretable directions by utilizing the so-called *Hessian penallty*, penalizing the norm of the off-diagonal elements of the Hessian matrix of a network. However, in our implementation, we were not able to obtain convincing results; we plan to analyze in the future with the authors' implementation when released.

### 3.2 LEARNING DISENTANGLED REPRESENTATIONS

We now discuss our approach to learning disentangled representations of a real dataset $X$. We start by training a GAN on $X$, and finding a set of $k$ interpretable orthogonal directions of unit length. We stack them vertically into a matrix $\boldsymbol{N} \in \mathbb{R}^{D \times k}$. In several methods (**DS**, **CF**, **GS**) the obtained directions are already orthogonal; other methods can be augmented with this constraint by performing the QR projection (Peebles et al., 2020) or parametrizing $\boldsymbol{N}$ via the matrix exponential (Voynov & Babenko, 2020). The obtained directions span a "disentangled subspace" of $\mathcal{W}$, with the basis formed by the columns of $\boldsymbol{N}$. By projecting a latent code $\mathbf{w}$ onto this space and finding its components in this basis, we obtain a new representation of the latent code, where ideally each coordinate rep-

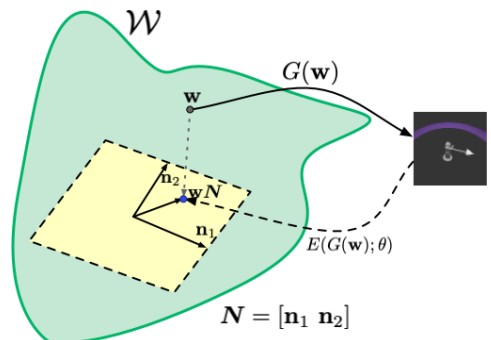

Figure 1: A visualization of our algorithm in the case of two interpretable directions $\mathbf{n}_1$ and $\mathbf{n}_2$.

resents an individual factor of variation. The resulting $k$-dimensional representation can be easily computed as $\mathbf{w}\boldsymbol{N}$, where we utilized orthogonality of $\boldsymbol{N}$. Note that this procedure is equivalent to first completing $\boldsymbol{N}$ to an arbitrary orthogonal basis in the entire space $\mathbb{R}^D$, and after the change of coordinates omitting all but the first $k$ informative components, in the spirit of the PCA projection.

To obtain such representations for *real data*, we perform the following step.
We start by constructing a large *synthetic dataset* $X_{gen} = \{\boldsymbol{w}_i, G(\boldsymbol{w}_i)\}_{i=1}^N$, with $\boldsymbol{w}_i$ being style vectors. The $k$-dimensional disentangled codes representing the images $G(\boldsymbol{w}_i)$ are then computed as $\boldsymbol{w}_i\boldsymbol{N}$. We now train an *encoder* network $E(\boldsymbol{x}; \theta) : \mathbb{R}^{C \times H \times W} \to \mathbb{R}^k$ by minimizing the following loss function:

$$\mathcal{L}(\theta) = \mathbb{E}_{X_{gen}} \|E(G(\boldsymbol{w}_i); \theta) - \boldsymbol{w}_i\boldsymbol{N}\|^2. \tag{3}$$

This approach is similar in spirit to generator inversion (Abdal et al., 2019; Zhu et al., 2020; Creswell & Bharath, 2018; Zhu et al., 2016), which is known to be a challenging problem and typically requires sophisticated algorithms. In our experiments, however, we were able to train encoders reasonably well without any particular tweaks, probably due to the fact that the modified latent codes $\boldsymbol{w}\boldsymbol{N}$ represent informative image attributes that are easier to be inferred.

**Hyperparameters:** to train the encoder, we need to fix the network architecture and training parameters; it is also affected by the random seed for initialization. We also need to choose the value of $N$ and sample $N$ training points.

**Summary.** Let us briefly summarize the proposed procedure to obtain disentangled representations of a dataset.

1. Train a non-disentangled GAN generator $G$ on the dataset.

2. Obtain a set of $k$ interpretable orthogonal directions of unit length in the latent space with one of the previously described methods; assume that they are arranged in a matrix $\boldsymbol{N} \in \mathbb{R}^{D \times k}$.

3. Train an encoder $E$ on synthetic data to predict the mapping $G(\boldsymbol{w}) \to \boldsymbol{w}\boldsymbol{N}$.

## 4 EXPERIMENTS

In this section, we extensively evaluate the proposed paradigm in order to assess its quality and stability with respect to various method hyperparameters and stochasticity sources. To achieve this, we perform an extensive sweep of random seeds and controllable generation methods and evaluate the obtained encoders with respect to multiple metrics. All our code and models are available at https://bit.ly/3ipb6dW.

## 4.1 Experimental setup

**Datasets.** We consider the following standard datasets: `3D Shapes` consisting of $480,000$ images with 6 factors of variations (Burgess & Kim, 2018), `MPI3D` consisting of $1,036,800$ images with 7 factors (Gondal et al., 2019) (more specifically, we use the `toy` part of the dataset), `Cars3D` – $17,568$ images with 3 factors (Fidler et al., 2012; Reed et al., 2015); we resize all images to $64 \times 64$ resolution. We also study the recently proposed `Isaac3D` dataset (Nie et al., 2020) containing $737,280$ images with 9 factors of variations; images are resized to $128 \times 128$ resolution.

**Model.** We use the recently proposed StyleGAN 2 model (Karras et al., 2020) and its open-source implementation in `Pytorch` from github[2]. Importantly, we **fix** the architecture and only vary the random seed when training models. For all datasets, we use the same architecture with $512$ filters in each convolutional layer and the style network with 3 FC layers. The latter value was chosen based on experiments in Nie et al. (2020). For `Isaac3D`, we perform a more of a proof-of-concept experiment by training a single GAN model and varying only random seeds and hyperparameters when training encoders. We employ truncation with a scale of $0.7$ for `Isaac3D` and $0.8$ for other datasets; we did not tune these values and selected them initially based on the idea that more realistic looking samples are beneficial for training the encoder, and the fact that `Isaac3D` is a more challenging dataset. In Appendices A and B we provide specific values of remaining hyperparameters, architecture and optimization details.

**Disentaglement methods.** We consider the four previously discussed methods, namely, **CF**, **GS**, **LD** and **DS**. For a fair comparison with VAEs in Locatello et al. (2019b;a); van Steenkiste et al. (2019), we use $k = 10$ for each method, i.e., we learn $10$–dimensional representations of data. We use the following hyperparameters for each method.

- **GS**: We fix $N = 20,000$ and sweep across random seeds for sampling.
- **LD**: We use the authors' implementation available at github[3] with default hyperparameters and backbone; we train it for $5,000$ iterations and sweep across random seeds.
- **DS**: We consider the outputs of first convolutional layers at resolutions $32$ and $64$, and the output of the generator; we average the results obtained for each of these layers. For the base point, we decided to simply fix it to the style vector $\boldsymbol{w}_0$ corresponding to $\boldsymbol{0} \in \mathcal{Z}$.

Recall that **CF** does not require any hyperparameters.
As a separate minor experiment, we provide examples of interesting directions found with our **DS** method in latent spaces of various high–resolution StyleGAN 2 models in Appendix E.

**Encoders.** For each set of directions discovered by each method, we train the encoder model as described in Section 3. For the first set of datasets, we use the same four–block CNN considered in Locatello et al. (2019b); specific details are provided in Appendix A. For `Isaac3D`, we consider the ResNet18 backbone (He et al., 2016), followed by the same FC net as in the previous case. We use $500,000$ generated points as the train set and sweep across random seeds.

**Disentanglement metrics.** We compute the following metrics commonly used for evaluating the disentanglement representations learned by VAEs: *Modularity* (Ridgeway & Mozer, 2018) and *Mutual information gap (MIG)* (Chen et al., 2018). We adapt the implementation of the aforementioned metrics made by the authors of Locatello et al. (2019b) and released at github[4]. We use $10,000$ points for computing the Mutual Information matrix.
Modularity measures whether each code of a learned representation depends only on one factor of variation by computing their mutual information. MIG computes the average normalized difference between the top two entries of the pairwise mutual information matrix for each factor of variation.

**Abstract reasoning.** Motivated by large-scale experiments conducted in van Steenkiste et al. (2019), we also evaluate our method on the task of *abstract reasoning*.

---

[2]https://github.com/rosinality/stylegan2-pytorch
[3]https://github.com/anvoynov/GANLatentDiscovery
[4]https://github.com/google-research/disentanglement_lib

In an abstract reasoning task, a learner is expected to distinguish abstract relations to subsequently re-apply it to another setting. More specifically, this task consists of completing a $3 \times 3$ *context panel* by choosing its last element from $3 \times 2$ *answer panel* including one right and five wrong answers forming Raven's Progressive Matrices (RPMs) (Raven, 1941).

We conduct these experiments on the `3D Shapes` dataset and use the same procedure as in van Steenkiste et al. (2019) to generate difficult task panels. An example of such a task panel is depicted in Figure 4. For this experiment we utilize the the open-source[5] implementation of Wild Relation Network (WReN) (Santoro et al., 2018) with default hyperparameters. The encoder is frozen and produces 10-dimensional representations, which in turn are fed into WReN.

**Fairness.** Another downstream task, which could benefit from disentangled representation, is learning fair predictions (Locatello et al., 2019a). Machine learning models inherit specifics of data, which could be collected in such a way that it can be biased towards sensitive groups causing discrimination of any type (Dwork et al., 2012; Zliobaite, 2015; Hardt et al., 2016; Zafar et al., 2017; Kusner et al., 2017; Kilbertus et al., 2017). Similarly to Locatello et al. (2019a), we evaluate *unfairness score* of learned representations, which is the average total variation of predictions made on data with the perturbed so-called *sensitive factor* value.

**Random seeds.** For each of the first three datasets, we train **eight** GANs by only varying the initial random seed; for `Isaac3D` we train a single model. For each generator, we then evaluate each method for **five** initial random seeds.

### 4.2 KEY EXPERIMENTAL RESULTS

Our key results are summarized in Figures 2 and 3 and Table 1. For each dataset and each method, we report the Modularity and MIG scores obtained using our approach. We compare our results with the results in the large scale study of disentanglement in VAEs (Locatello et al., 2019b, Figure 13). Note that Locatello et al. (2019b) evaluate various models by fixing all the architecture and optimization details and only varying a regularization strength for each method, as well as random seeds. Thus, the setup is close to ours. Results are provided there in the form of violing plots; the actual numerical values are available only for `Cars3D`. Detailed comparison on this dataset is provided at Figure 3. We observe that all the methods are able to achieve disentanglement scores

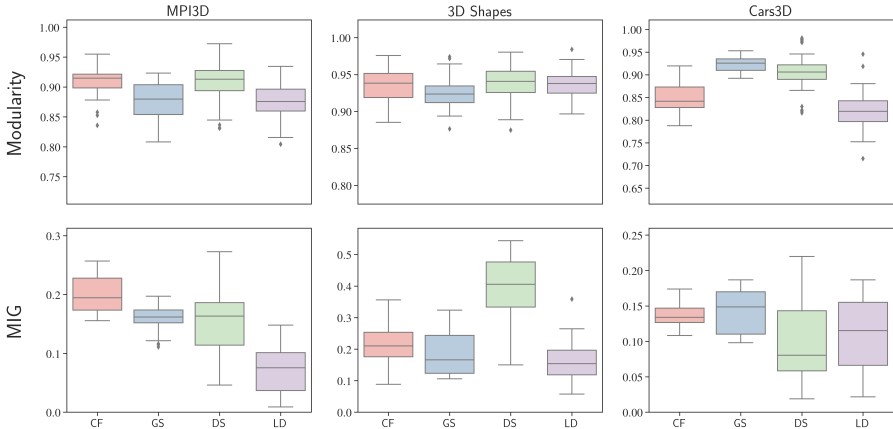

Figure 2: Modularity and MIG scores (higher is better) obtained for various encoders and datasets trained via the two-stage procedure as described in Section 3 for StyleGAN 2. We observe that a) average results are on par or outperform most of the VAE-based models (Locatello et al., 2019b) b), on the other hand, for many methods, our approach provides smaller variance; the variance is due to random seeds in generators and encoders, see Section 3.

competitive with the scores reported for VAE-based approaches, see Locatello et al. (2019b, Figure 13). E.g., on `Cars3D`, the average score of **CF** in terms of MIG exceeds the highest average

---

[5]https://github.com/Fen9/WReN

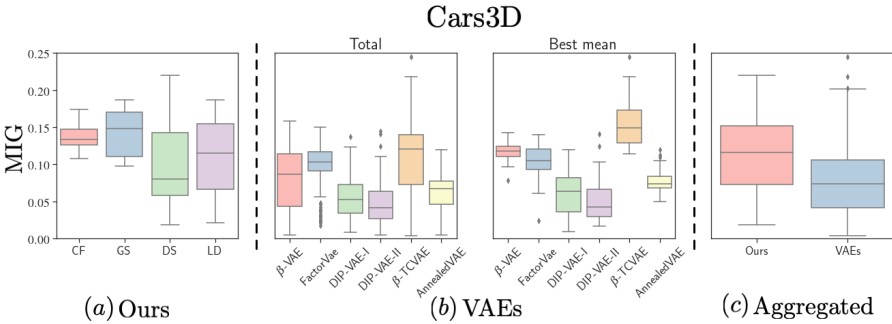

(a) Ours  (b) VAEs  (c) Aggregated

Figure 3: Comparison of our MIG scores on `Cars3D` with various VAE based approaches; numbers are provided by the authors of Locatello et al. (2019b). **Total** denotes the entire distribution of VAE scores with respect to the hyperparameter/seed space; For **Best mean** for each VAE–based method, we select a **single best** hyperparameter according to the mean (over random seeds) MIG value and plot the distribution only with respect to seeds. For (c), we plot the distributions of MIG over all configurations (method, seed, hyperparameters).

| Method | Dataset | | | |
|---|---|---|---|---|
| | Cars3D | 3D Shapes | MPI3D | Isaac3D |
| **GS** | **0.143** $\pm$**0.029** | 0.188 $\pm$0.070 | 0.160 $\pm$0.022 | 0.190 $\pm$0.01 |
| **CF** | 0.137 $\pm$0.017 | 0.216 $\pm$0.073 | **0.198** $\pm$**0.030** | **0.448** $\pm$**0.005** |
| **DS** | 0.098 $\pm$0.058 | **0.398** $\pm$**0.086** | 0.155 $\pm$0.053 | 0.352 $\pm$0.07 |
| **LD** | 0.107 $\pm$0.050 | 0.160 $\pm$0.061 | 0.073 $\pm$0.039 | 0.140 $\pm$0.043 |
| InfoStyleGAN | - | - | - | 0.328 $\pm$0.057 |
| InfoStyleGAN* | - | - | - | 0.404 $\pm$0.085 |

Table 1: We provide mean and standard devations of MIG for each method and for each dataset. InfoStyleGAN and InfoStyleGAN* correspond to models of various capacity (large and small) as specified in (Nie et al., 2020). For the first three datasets randomness is due to random seed both in generators and encoders; for `Isaac3D` the generator is fixed and we only vary the random seed and hyperparameters when training encoders.

score for all the competitors. Notice that the variance due to randomness tends to be smaller than for VAEs, and we are able to consistently obtain competitive disentanglement quality. **CF** and **GS** tend to be more stable on average, while **DS** is capable of achieving higher scores but is less stable. In many cases, VAEs underperform for a large portion of the hyperparameter/seed space. Figure 3 (c) also provides the distributions of MIG values aggregated over random seeds, hyperparameter values and methods, demonstrating that scores achieved by the post-hoc disentanglement are generally higher.

While the `MPI3D` was not studied in Locatello et al. (2019b), we note that our MIG values are comparable with carefully tuned VAE models achieving the best results in "'NeurIPS 2019 disentanglement challenge'"[6]. We also note that our **DS** method performs reasonably well by achieving the highest possible results on all the datasets and the best average result on `3D Shapes`. It appears that **LD** struggled to reliably uncover factors of variations. One possible reason is that we searched only for 10 directions, while unlike other methods, it does not have an appealing property allowing us to select *top k* directions with respect to some value, e.g., as in the PCA case. A possible solution to that might be discovering a new approach of the unsupervised selection of the best directions from a large set of candidates.

In Table 1 we also provide our results for `Isaac3D`. Interestingly, with the **CF** method, we are able to achieve MIG competitive with InfoStyleGAN* and outperform InfoStyleGAN. This suggests that auxiliary regularizers *may not be necessary*, and the latent space of StyleGANs is already disentangled to a high degree.

For the methods achieving the best results in terms of MIG, we provide the corresponding Mutual Information matrix in Appendix D and visualize latent traversals in Appendix C.

---

[6]https://www.aicrowd.com/challenges/neurips-2019-disentanglement-challenge

### 4.2.1 ABSTRACT REASONING AND FAIRNESS

We now verify whether the learned representations can be utilized for abstract reasoning tasks. We also verify the fairness of these representations, as previously discussed. See Figures 4 and 5 for the results. Note that our method allows for training abstract reasoning models with consistently high accuracy (e.g., mostly exceeding 95% in the case of **CF**). These results are competitive with VAE-based models van Steenkiste et al. (2019, Figure 11); however, the difference is hard to estimate quantitatively as the numerical results are not provided there. Similarly, we find that in terms of *unfairness*, our method finds the representations with the distribution of scores comparable to those produced by VAEs, see Locatello et al. (2019a, Figure 2); however, the variance for our methods is smaller in all the cases. On average, the VAE methods are slightly better on `3D Shapes` and slightly worse on `Cars3D`.

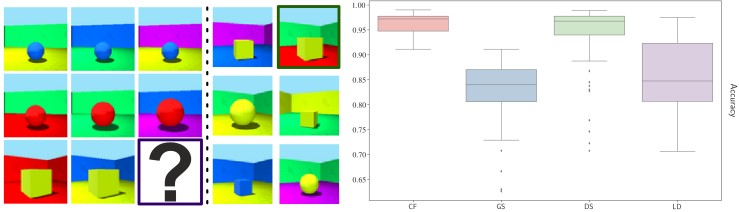

Figure 4: *(Left)* An example of the *abstract reasoning* task. The goal of the learner is to correctly choose the correct answer (marked with green in this example) from the *answer* panel, given the *context* panel. *(Right)* Accuracy obtained by training WReN with the (frozen) encoders obtained using one of the discussed methods. In most of the cases, we reliably obtain a sufficiently high accuracy value.

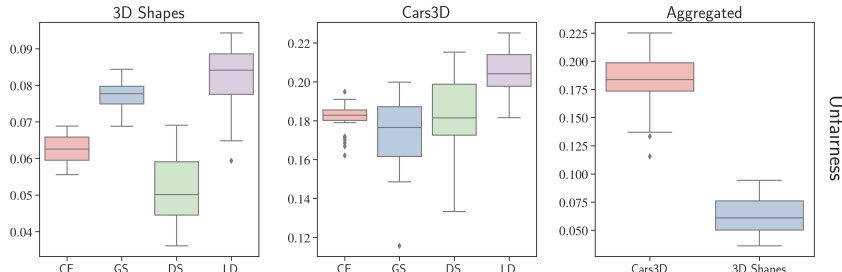

Figure 5: Distribution of unfairness scores (the lower, the better); we can observe that the scores are relatively low despite different hyperparameters and random seed setups.

**Experiments on non-style-based GANs.** The main bulk of our experiments was conducted using StyleGANs. As a proof of concept, we verify whether our method works for other non-style-based GANs, specifically, we consider ProGAN (Karras et al., 2018); all the details and experimental results are covered in Appendix F. We compare our `DeepSpectral` method to `ClosedForm` and `LatentDiscovery` since these methods are the easiest to extend to non style-based GANs. The obtained results are slightly inferior to StyleGAN 2, and we attribute this behavior to the larger gap between real data and ProGAN samples compared to StyleGAN 2 samples. Since the encoder in our scheme is trained on the synthetic data, the quality of samples is crucial.

## 5 CONCLUSION

In this work, we proposed a new unsupervised approach to building disentangled representations of data. In a large scale experimental study, we analyzed many recently proposed controlled generation techniques and showed that: (i) Our approach allows for achieving disentanglement competitive with other state-of-the-art methods. (ii) We essentially get rid of critical hyperparameters, which may obstruct obtaining high quality disentangled representations in practice. A number of open questions, however, still remains. Firstly, the existence of directions in the GAN latent space *almost perfectly* correlated with exactly one of the factors of variations is quite surprising and requires further theoretical understanding. Additionally, there has been some evidence that linear shifts may perform subpar compared to more intricate non-linear deformations in a modified latent space. We leave this analysis for future work.

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

## A  ARCHITECTURES

Here, we provide the model hyperparameters used in our study.

| Mapping Network |
|---|
| (FC × `n_mlp`) `latent_dim` × `latent_dim` |
| **Synthesis Network** |
| (4×4 Conv) 3×3× `width` × `width` |
| (4×4 Conv) 3×3× `width` × `width` |
| (8×8 Conv) 3×3× `width` × `width` |
| (8×8 Conv) 3×3× `width` × `width` |
| (16×16 Conv) 3×3× `width` × `width` |
| (16×16 Conv) 3×3× `width` × `width` |
| (32×32 Conv) 3×3× `width` × `width` |
| (32×32 Conv) 3×3× `width` × `width` |
| (64×64 Conv) 3×3 × `width` × `width` |
| (64×64 Conv) 3×3× `width` × `width` |

(a) Generator

| |
|---|
| (64×64 Conv) 3×3× `width` × `width` |
| (64×64 Conv) 3×3× `width` × `width` |
| (32×32 Conv) 3×3×`width`×`width` |
| (32×32 Conv) 3×3×`width`×`width` |
| (16×16 Conv) 3×3×`width`×`width` |
| (16×16 Conv) 3×3×`width`×`width` |
| (8×8 Conv) 3×3×`width`×`width` |
| (8×8 Conv) 3×3×`width`×`width` |
| (4×4 Conv) 3×3 × `width` × `width` |
| (4×4 Conv) 3×3 × (`width` + 1) × `width` |
| (4×4 FC) (16 ·`width`) × `width` |
| (4×4 FC) `width`×1 |

(b) Discriminator

Table 2: Generator and discriminator architectures for the StyleGAN 2 models for generating the image of resolution $128 \times 128$. "FC ×`n_mlp`" denotes `n_mlp` dense layers; "$2^k \times 2^k$ Conv" denotes the convolutional layers in the $2^k$ resolution block. For resolution $64 \times 64$, the first block in the discriminator and the last block in the generator are omitted.

| |
|---|
| Conv $4 \times 4 \times 3 \times 32$, `stride=2` |
| ReLU |
| Conv $4 \times 4 \times 32 \times 32$, `stride=2` |
| ReLU |
| Conv $4 \times 4 \times 32 \times 64$, `stride=2` |
| ReLU |
| Conv $4 \times 4 \times 64 \times 64$, `stride=2` |
| ReLU |
| FC 1024 × `f_size` |
| ReLU |
| FC `f_size` × 10 |

Table 3: Encoder architecture used in our experiments. `f_size` is 256 for `3D Shapes` and `MPI3D`, and 512 for `Cars3D` and `Isaac3D`. For `Isaac3D` the convolutional block is replaced with ResNet18 without the last classification layer.

# B  HYPERPARAMETERS

Here, we provide the training hyperparameters for StyleGAN 2, encoders and WReN.

| Parameter | Value |
|---|---|
| iter | 200000 for Isaac3D and 300000 for other datasets |
| batch | 32 |
| n_sample | 64 |
| size | 128 for Isaac3D and 64 for other datasets |
| r1 | 10 |
| path_regularize | 2 |
| path_batch_shrink | 2 |
| d_reg_every | 16 |
| g_reg_every | 4 |
| mixing | 0.9 |
| lr | 0.002 |
| augment | False |
| augment_p | 0 |
| ada_target | 0.6 |
| ada_length | 500000 |
| latent_dim | 512 |
| n_mlp | 3 |
| width | 512 |
| truncation | 0.7 for Isaac3D and 0.8 for other datasets |
| mean_latent | 4096 |
| input_is_latent | True |
| randomize_noize | False (for evaluation) |

Table 4: Training hyperparameters of the StyleGAN 2 model. Our implementation is based on https://github.com/rosinality/stylegan2-pytorch; we modified it to pass the number of filters (width) for convolutional layers and the latent dimension as hyperparameters.

Table 5: Training hyperparameters for encoders. Table 6: Training hyperparameters for WReN when solving the abstract reasoning tasks.

| Parameter | Value |
|---|---|
| Batch size | 128 |
| Optimizer | Adam |
| Adam: beta1 | 0.9 |
| Adam: beta2 | 0.999 |
| Adam: epsilon | $1e-8$ |
| Adam: learning rate | 0.001 |
| Training epochs | 20 |
| Learning rate decay: step size | 10 |
| Learning rate decay: gamma | 0.5 |
| Latent space dim: | 10 |

| Parameter | Value |
|---|---|
| Batch size | 32 |
| Optimizer | Adam |
| Adam: beta1 | 0.9 |
| Adam: beta2 | 0.999 |
| Adam: epsilon | $1e-8$ |
| Adam: learning rate | 0.0001 |
| Training steps | 100000 |
| Learning rate decay: step size | 10 |
| Learning rate decay: gamma | 0.5 |

## C  LATENT TRAVERSALS

In this section, we visualize traversals of the latent space for the best model (in terms of MIG) on each dataset. Best viewed in color and zoomed.

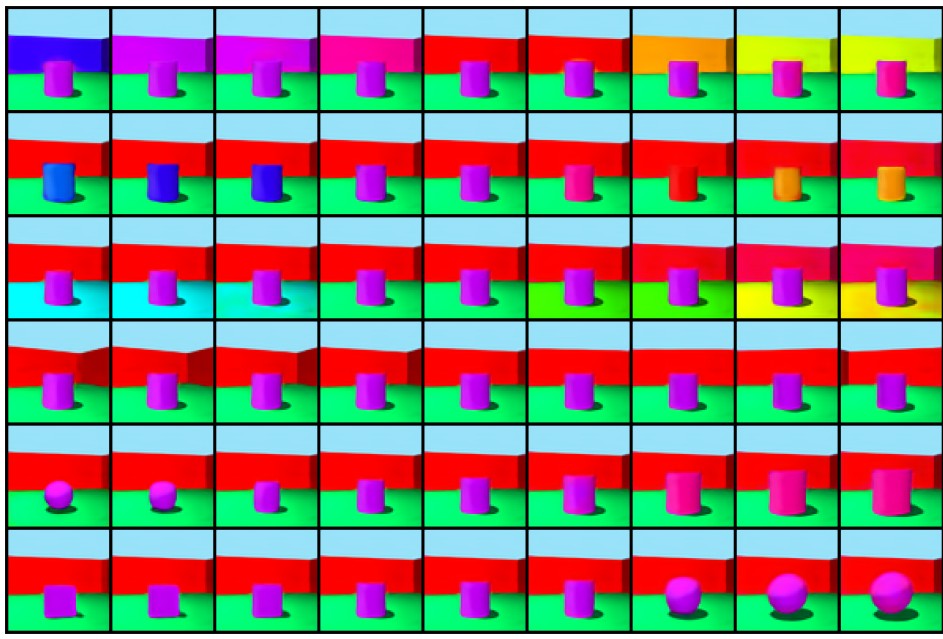

Figure 6: Latent space traversal for `3D Shapes`. We observe that all directions are almost perfectly disentangled, except for *shape* (6th row) and *scale* (5th row).

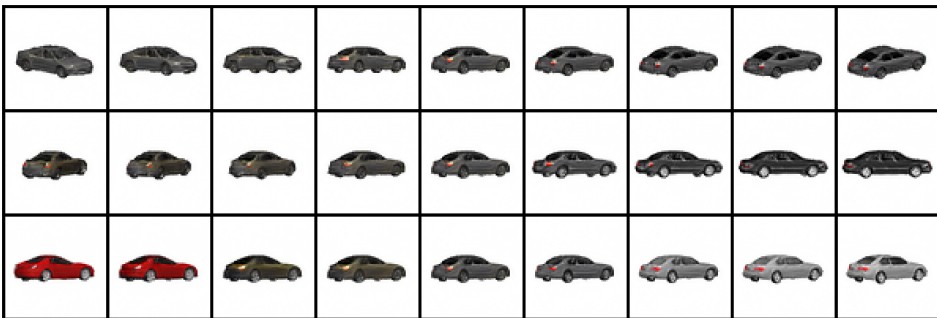

Figure 7: Latent space traversal for `Cars3D`. In principle, all the factors of variation (two rotations and car model) were captured, however, we can observe some entaglement.

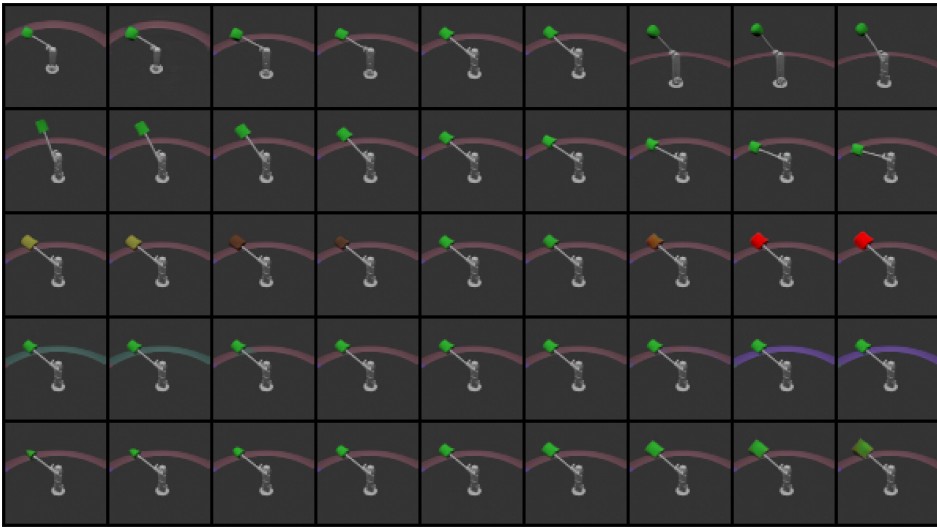

Figure 8: Latent space traversal for `MPI3D`. We observe that samples are of excellent visual quality, and found directions are reasonably disentangled, except for *shape* and *scale* (5th row), and *camera height* and *shape* (1st row).

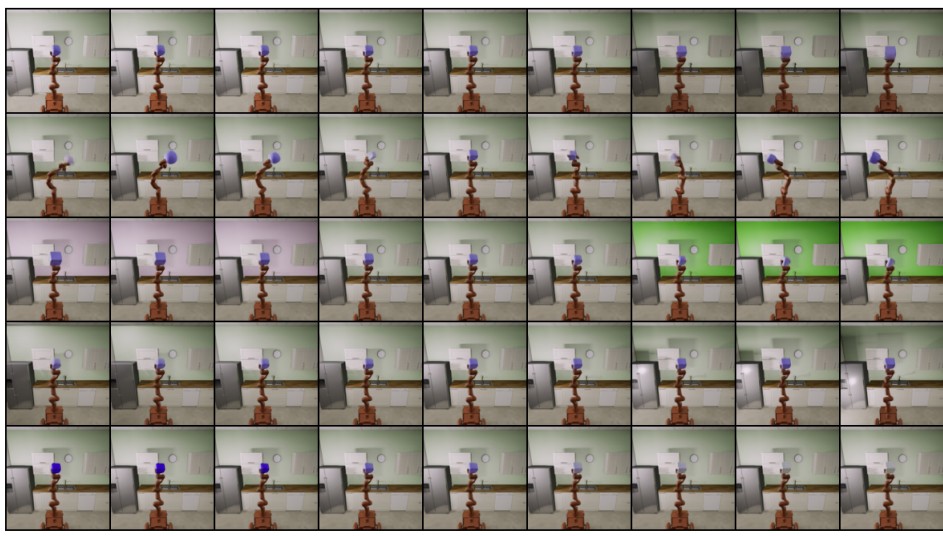

Figure 9: Latent space traversals for `Isaac3D` (selected directions). It appears that texture variations, e.g., lighting, color, shape are easier to be inferred than 'physical' directions, such as the robot handle rotations.

# D    WHICH FACTORS MODELS LEARN?

Let us now briefly analyze what factors of variations seem to be the easiest for our models to infer. For this we plot the Mutual Information matrices obtained for the best of our models. It seems that the texture-based factors are easier to learn, while the discovery of physical features is more challenging.

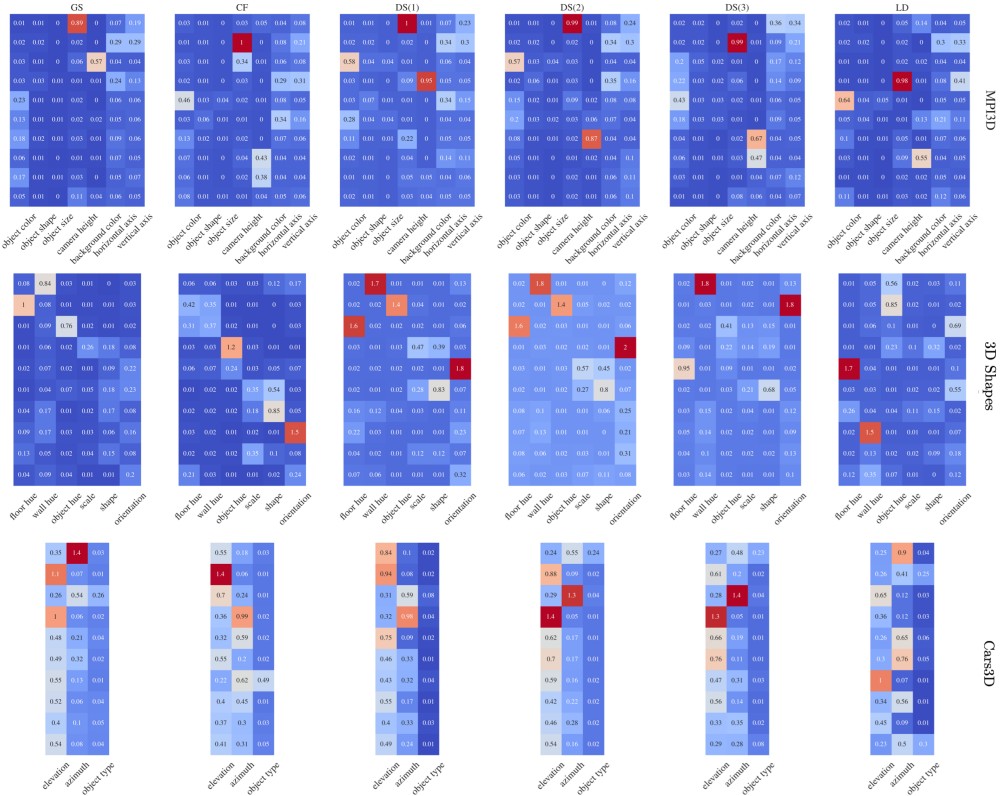

Figure 10: Mutual information matrices for various methods and datasets obtained for the model with highest overall MIG score; a higher value indicates stronger mutual information.

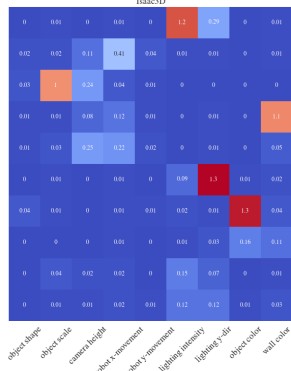

Figure 11: Mutual information matrix for the best method on the Isaac3D dataset.

# E    LATENT TRAVERSALS FOR LARGE SCALE MODELS

In this section, we visually inspect the interesting directions found by our method on the high–resolution GAN models. We use checkpoints available at `https://github.com/justinpinkney/awesome-pretrained-stylegan2`; from top to bottom the datasets are: `LSUN Church`, `FFHQ`, `Anime portraits`.

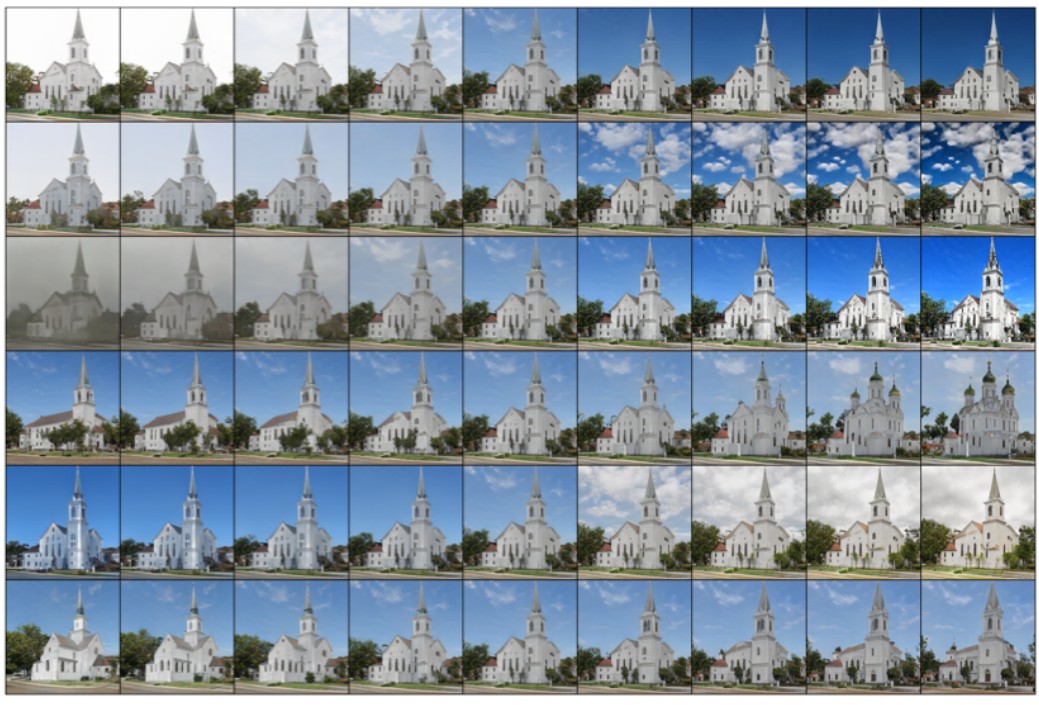

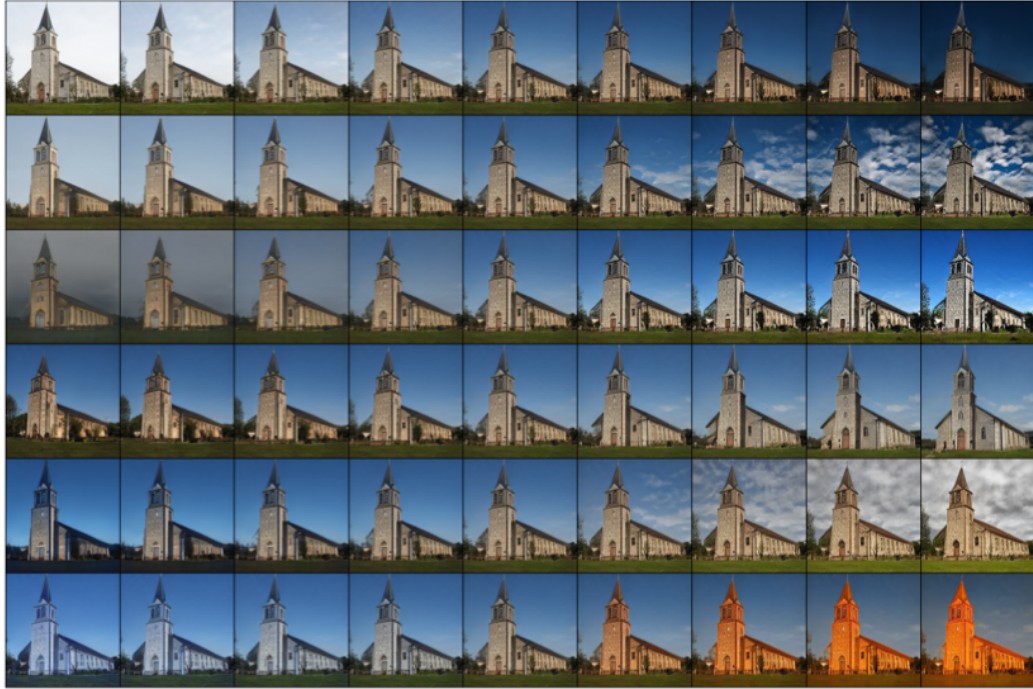

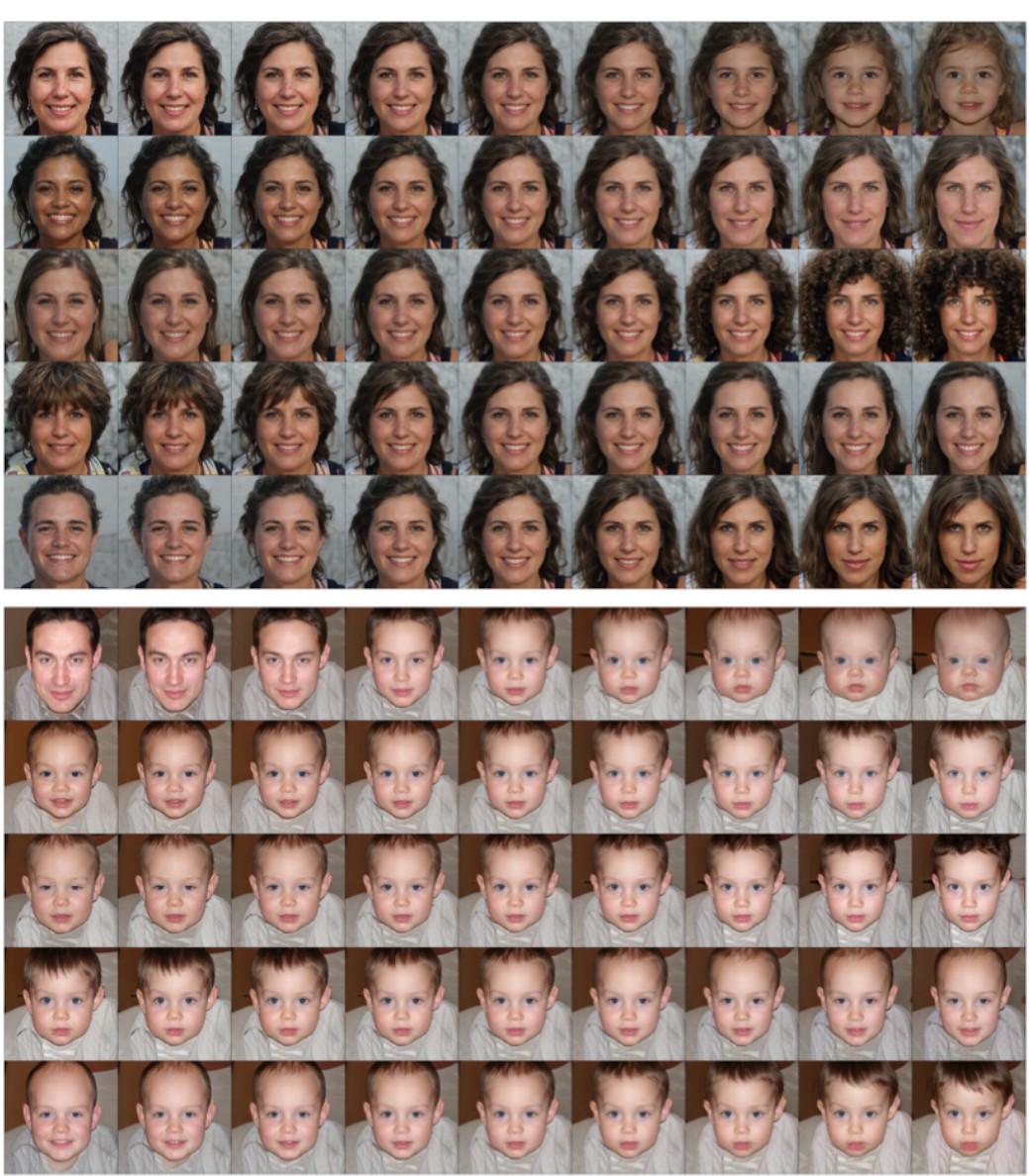

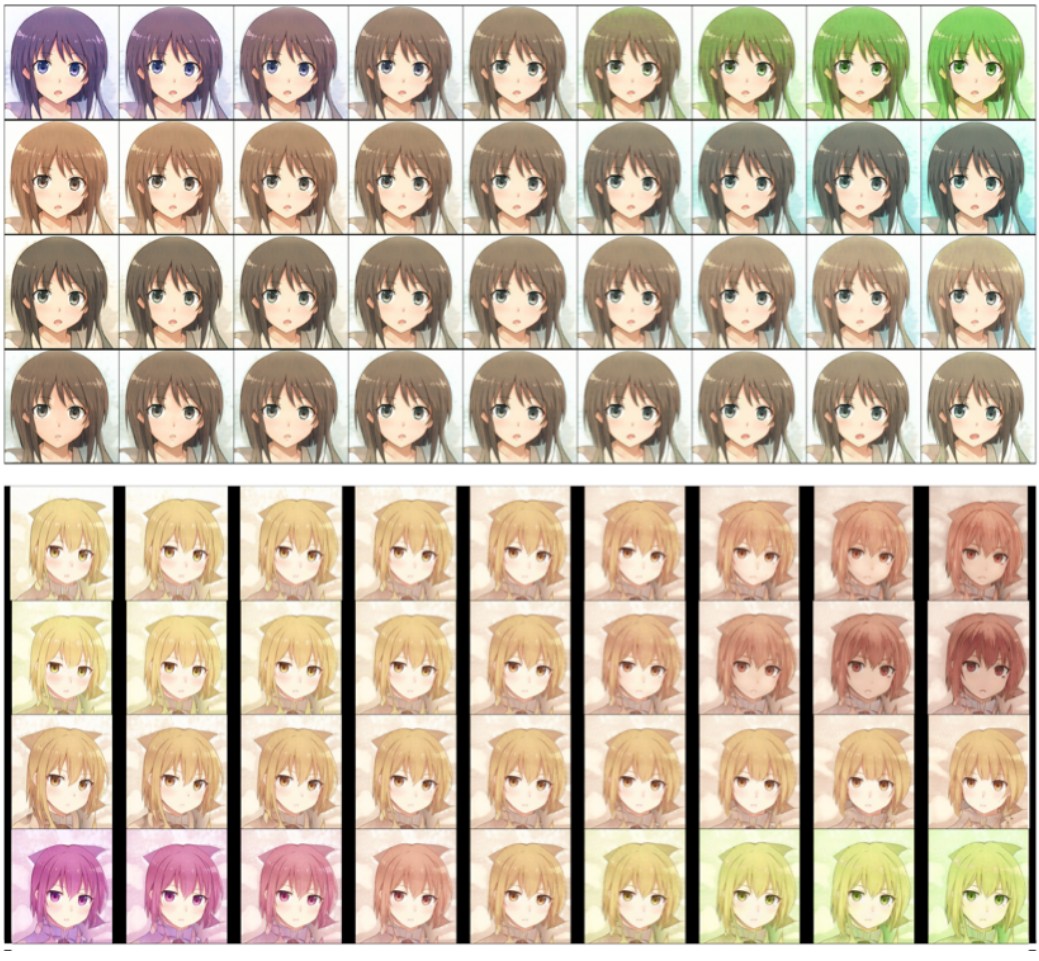

# F    EXPERIMENTS ON PROGAN

In this section we briefly describe our experiments with the non-style based GAN model, namely, we study ProGAN. We conduct our experiments on the 3D Shapes dataset. We used the code available at https://github.com/akanimax/pro_gan_pytorch. We trained the default model for 2 epochs at resolutions $8, 16, 32$ and for 4 epochs at resolution $64$ (this was sufficient for model to converge since the dataset is extremely large); we used batch size $128$. We consider three methods, most easily adapted to the non-style generators: **CF**, **DS** and **LD**. For **CF** we used the first ConvTranspose2d layer, and for **DS** we considered the outputs of convolutional blocks at resolutions $16, 32$ and $64$. **LD** works with this setup out of the box. Results are provided in Table 7. We see that the best obtained MIG score for this model roughly matches the average result of the **CF**, **GS** and **LD** methods for StyleGAN 2 models, however, is outperformed by the **DS** in many cases.

| Method | Metrics | |
| --- | --- | --- |
| | MIG | Modularity |
| **DS** | $\mathbf{0.138} \pm_{\mathbf{0.009}}$ | $0.900 \pm_{0.018}$ |
| **CF** | $0.110 \pm_{0.009}$ | $0.927 \pm_{0.017}$ |
| **LD** | $0.106 \pm_{0.026}$ | $\mathbf{0.940} \pm_{\mathbf{0.032}}$ |

Table 7: The experimental results for 3D Shapes dataset with the ProGAN model.

