# OpenReview forum: "On Disentangled Representations Extracted from Pretrained GANs"
_ICLR.cc/2021/Conference — Reject_

### Official Review · AnonReviewer4 · 2020-10-28
**Potentially a useful contribution, but provided experiments are incomplete and have several other issues**

**Rating:** 4
**Confidence:** 4

**Review:**

### Summary

This paper proposes a non-conventional approach to learning disentangled representations through finding interpretable directions in the latent space of a pre-trained Style-GAN (Karras et al., 2018). To find these directions, several existing techniques from the recent literature on controllable image generation are considered. To apply this technique to existing (real) images, an encoder network is trained on the generated images to learn a mapping  from images to disentangled representations.

As the main contribution it is shown how this approach allows one to recover disentangled representations on several standard datasets, which perform similar to conventional techniques for learning disentangled representations based on VAEs. Nonetheless, the paper argues that this approach is preferred since it does not rely on hyper-parameter tuning using ground-truth factors, which is typically necessary for conventional approaches (Locatello et al., 2018). Finally, the paper also contributes its own approach for discovering interpretable latent directions, but which is a simple heuristic derived from the “ClosedForm” method proposed by Shen & Zhou (2020).

### Pro’s / Con’s / Justification

The paper is reasonably well written, although I should note that it violates the required style guide for formatting paragraphs. By my count this paper would otherwise have exceeded the page limit of 8 pages by about 22 lines and I have previously reached out to the AC about this.

I am somewhat conflicted about the significance of the contribution. On the one hand, I like the approach as an alternative to VAE-based approaches and I think that it is valuable to connect the field of disentangled representation learning with the field of controllable image generation. On the other hand, the insight that there is a correspondence between interpretable latent directions and disentangled factors of variation is not new as is evident from the recent literature cited here. Indeed, I would argue that the main contribution is simply the application of these techniques to a standard benchmark for learning disentangled representations. Here I also note that the proposed approach for going from disentangled representations for generated images to disentangled representations for real image is trivial. Nonetheless I would still consider the application itself to be a significant contribution. However, the current experiments are incomplete and have several other issues, which I will comment on next.

First, I note that while the paper claims to investigate an alternative route to learning disentangled representations based on “state-of-the-art GANs”, only two variations of Style-GAN are considered. This is problematic, since it leaves it unclear to what extent the evaluated approach applies to other GANs. For example, were it found that the same techniques do not yield disentangled representations for other GANs, one would end up in the very same situation as when using conventional (VAE-based) approaches, namely by having to use ground-truth factors of variation to perform model-selection. Since this is the main argument as to why the proposed approach is preferred it is vital that this is investigated.

Second, and related to this, is the quality of the provided comparison to VAE-based approaches. The reader is required to compare the boxplot in Figure 1 to the violin plots in Figure 13 in Locatello et al. (2018). Other than this being far from ideal and only allowing for a rough comparison, it is not clear to me that these results can be compared since Locatello et al. (2018) reports an average that additionally includes six different regularization strengths for the VAE-based methods. While it can be argued that this is a limitation of the VAE-based approach (i.e. having to consider multiple regularization strengths) it is not clear to what extend the style-based generators, the choice of encoder, and the hyper-parameters of the various techniques for discovering latent directions considered (such as the number of latent directions to search for) here are already “optimal” for these problems (in which case they should also be varied to allow for a fair comparison). Moreover, the results for “DeepSpectral” are reported separately for each choice of layer, while they should be aggregated into a single score to allow for a fair comparison. I note that in that case, the results don’t look all that convincing. “GS” works well on some datasets, but worse on others, “DS” has large variance on all datasets and performs average, “CF” has large variance and mixed results for MIG across data sets, and “LD” generally tends to perform bad. Can it really be argued that the GAN-based approach is more stable and does not require model-selection from these results? Certainly there is no clear best method for interpretable latent discovery. It is also not clear how the size of the interval for the regularization parameter in Locatello et al. (2018) compares to that explored for DS. To facilitate a better comparison I would argue that a comparison of various VAEs having the _optimal_ regularization strength, to the reported results when using pre-trained GANs is needed. I would also like to see the authors aggregate results across methods for interpretable latent discovery and compare that to an aggregate over possible VAEs (and regularization strengths). Based on the reported results I would expect the variance to be equally large in both cases, which would leave it unclear whether this pre-trained GAN-based approach is preferred.

My final concern regarding the experimental evaluation is concerned with the results presented in section 4.2.1, where the learned representations using the approach presented here are applied to abstract visual reasoning tasks and to a fairness task. These tasks were used to demonstrate the benefits of representations that are disentangled, but here they are used to validate the learned representation. For example, Figure 2 is compared to Figure 11 in van Steenkiste et al. (2019) and it is observed that the representations acquired from pre-trained GANs are better than VAE-based representations by comparing the spread. However, the “pre-trained” line in Figure 11 in van Steenkiste et al. (2019) only contains a random selection of representations learned by VAEs _without_ considering their disentanglement. Hence, it is not clear what can be concluded from this other than that more disentangled representations perform better on this task, but this was already observed in  van Steenkiste et al. (2019).

The remaining contribution that is “DeepSpectral” currently feels ad-hoc and more of a heuristic, even though there is a correspondence to GS in the linear case. One reason for this is that it performs about average in all cases and has relatively large variance (if averaging across the main hyper-parameter). Perhaps this part of the contribution can be strengthened by providing additional analysis, why this is a reasonable approach, i.e. by analyzing its behavior in the linear case where there is a correspondence. It could also be useful to see whether the number of layers can be chosen dynamically based on some internal statistic, but that does not require ground-truth disentanglement factors.


### Detailed comments

* The title is too broad and not very informative. Please narrow it down to using pre-training GANs and methods for latent direction discovery

* It is often claimed that by using the latent-discovery approach one gets rid of essentially all critical hyper-parameters. However, there is not enough evidence provided that the remaining hyper-parameter values chosen do not affect the quality of the discovered representations. For example, the base point in GS is always fixed. Similarly, for LD only default hyper-parameters are used and the effect of the regularization parameter is not explored.

* W is sometimes used to refer to weights (eg. in the first layer of the transformation) and other times to the latent space learned by style GAN from which interpretable directions are recovered. I think in the “ClosedForm” paper they use A in place of W for the network weights, which may be more clear.

* In general, please report the full experimental details used, even though they are the “default values”. For example, currently it is unclear to me how many training steps were sed to obtain the results in Figure 2.

* Please take a moment to go through the references and correctly cite papers that have been published.

### Post Rebuttal

I have read the other reviews and the author's response. I also had another look at the revision. While I appreciate the work that was put in the revision, my main concerns remain:

* This paper proposes an alternative to VAE-based approaches to learning disentangled representations, which are known to require hyper-parameter selection based on ground-truth factors (as pointed out in Locatello et al., 2019). Hence, to improve over VAEs in this regard, it is crucial that the proposed framework does not suffer from this same issue or least consistently performs better compared to VAEs. Unfortunately this is simply not the case: across Figure 2 and Table 1 there is no variation (among CF, GS, DS, LD) that consistently performs well or consistently outperforms VAEs. Similarly in Figure 3c it can be seen how the GAN-based approach and VAE-based approach perform similarly (especially if the outliers on the top would be included in the mean). Similar fluctuations can essentially be observed across all figures and which does not take into account yet the choice of GAN or other hyper-parameters that were kept fixed (as per my initial review). Further, notice how in Table 7 (when using ProGAN as opposed to StyleGAN), the best performing variation (DS) now performs more than 50% worse according to MIG -- which is not factored in the comparison to VAEs.
* The abstract visual reasoning experiment is flawed, since the comparison to van Steenkiste et al. (2019) figure 11 considers a random selection of pre-trained VAEs for which it is unclear whether they were disentangled or not.
* DS is a simple heuristic that generally does not outperform other approaches. Indeed, notice how not reporting the results for DS separate for each layer (as was done previously) now results in large fluctuations. In particular, for the four datasets considered in Figure 1 DS outperforms other methods only on 3D shapes. Further, although the authors in their rebuttal argue that DS is not a simple heuristic since they “build [DS] on the intuition that singular vectors of the Jacobian provide a set of locally ``independent directions with respect to the perceptual metric”, there is no evidence provided that this intuition is correct. I had suggested ways according to which this part of the contribution could be strengthened (i.e. by analyzing it in the linear case, or attempting to automatically select a good layer) but this was not further explored.

I do think that there is significant value in a systematic analysis of GAN-based approaches applied for disentanglement in this way, as it could serve as a useful benchmark for existing (VAE-based) approaches to compare against. However, the current set-up falls short at this as it is not sufficiently systematic and certain variations remain insufficiently unexplored (like other kinds of GANs). Further, this is not how this work is currently motivated in the paper or how the comparisons are performed.

These concerns are irrespective of whether the reader needs to perform a manual comparison to Locatello et. al (2019). Although I noted in my review that this is far from ideal, I can understand how this may be necessary if Locatello et al. (2019) can not provide a smaller range of hyper-parameters. In that sense, I believe that the authors did a good job incorporating VAE results at such short notice.

---

> ### Author Response · Authors · 2020-11-24
> **Response to AR4**
>
> We thank the reviewer for the review and address each of the comments below:
>
> [Alternative GAN architectures.] We chose StyleGAN 2 since it is the current state-of-the-art for an unconditional generation. For less powerful models, however, our approach also achieves decent performance. In Appendix F, we have added results for ProGAN [A] on the 3D Shapes dataset. The results are slightly inferior to StyleGAN2, and we attribute this behavior to the larger gap between real data and ProgGAN samples compared to StyleGAN2 samples. Since the encoder in our scheme is trained on the synthetic data, the quality of samples is essential; therefore, it is important to use the best available model. Overall, we argue that the choice of the GAN model for our approach does not require ground truth factors of variation, and the current SOTA GAN architecture established in the community can be used for a particular unsupervised disentanglement task.
>
> [Comparison to VAE.] We have added the direct comparison in Section 4.2 in the new version. For these plots, we used the pretrained models for Cars3D provided in https://github.com/google-research/disentanglement_lib.
> We asked the authors (Locatello et al., ICML’19) for exact numbers for other datasets, but they could not provide them. Unfortunately, we are not able to re-train thousands of models during the discussion phase.
>
> [the choice of encoder, and the hyper-parameters of the various techniques for discovering latent directions considered...should also be varied to allow for a fair comparison] We aimed to perform the evaluation in the setup, which is as comparable to the setup in (Locatello et al., ICML’19) as possible. Locatello et al. computed the variance only with respect to regularization strengths and random seeds, all other hyperparameters being fixed. These hyperparameters include the dimensionality of the latent space, the architecture, and the details of the training protocol, such as learning rate, batch size, etc. For this reason, we also considered these hyperparameters as fixed. Meanwhile, we sweep over random seeds, which appear in GAN training, discovering latent directions, and encoder training.
>
> [the results for “DeepSpectral” are reported separately for each choice of layer, while they should be aggregated into a single score to allow for a fair comparison] We agree with this point and report the aggregated results for DS in the new revision.
>
> [I would also like to see the authors aggregate results across methods for interpretable latent discovery and compare that to an aggregate over possible VAEs (and regularization strengths).] We provide the results of this experiment in Figure 3. In this case, the average scores obtained by our method are generally higher.
>
> [Сomparison of various VAEs having the optimal regularization strength] As proposed by the reviewer, we compare our approach with VAEs, given the optimal regularization strength. Note that in this case, the comparison is not completely fair since the knowledge of ground truth generative factors is used in VAEs. The results are provided in Figure 3, demonstrating that CF and GS can achieve competitive performance even in this setup.
>
> [Abstract visual reasoning tasks.] In this experiment, we verify if the representations produced by our approach can be used as alternative unsupervised representations to solve abstract reasoning tasks. Rather than using proxies (e.g., MIG, Modularity, etc.), we show that our representations result in more stable abstract reasoning performance.
>
> [Simplicity of DS] We respectfully disagree with the reviewer's comments on the DS's simplicity. While it indeed can be reduced to CF in a basic case, this does not imply that it is a “simple heuristic”. Conceptually, while the authors of CF argue that empirically the first layer of generators determines the values of important factors of variations, we build our method on the intuition that singular vectors of the Jacobian provide a set of locally ``independent directions with respect to the perceptual metric, which is the desired property of disentangled representations. From a computational point of view, our method additionally takes significantly more effort to implement since the efficient realization requires, e.g., Pearlmutter’s trick.
>
> [Changing the title] In the new revision, we have changed the title to be more specific.
>
> [Hyperparameters and other details] Note that all the hyperparameters and training settings for our setup were initially provided in Appendix B.
>
>
> [A] [Progressive growing of GANs for improved quality, stability, and variation](https://arxiv.org/abs/1710.10196)

---

### Official Review · AnonReviewer3 · 2020-10-29
**Interesting empirical observations, needs better writing**

**Rating:** 6
**Confidence:** 4

**Review:**

This paper proposes a method to learn disentangled representations. The idea is to use an existing GAN generator and use an existing controllable generation algorithm (such as ClosedForm or GANspace) to find a set of “important” directions in the latent space. These subspaces spanned by these directions could already represent disentangled factors of variation.  The main contribution of the paper is to propose learning a mapping from input to this subspace (that inverts the generator).

Pro:

The empirical evaluation shows that the unsupervised controlled generation methods are able to find a set of directions in the latent space that represent disentangled factors of variation and can achieve on-par performance on standard disentanglement benchmark datasets.

The experiments are fairly comprehensive (with multiple datasets, and several controllable generation methods)

Con:

The success of the approach depends on whether the original algorithm (such as Closedform or GANspace) is able to find disentangled subspaces. If I understand correctly the paper only proposes to invert the generation process from the disentangled subspace (which is learned by existing methods) to the image. This somewhat limits the novelty. (Though I think the empirical finding that existing controllable generation algorithms also work well for disentanglement is also interesting)

The writing seems fairly unpolished. For example, it took some effort to understand that the authors are defining a new basis with columns of A, and the vectors w_i are defined under the coordinate system of the basis. These should be formally defined and clearly stated to avoid ambiguity. As another example, it is unclear what “sources of randomness” means. Sometimes this refers to hyper-parameters, while other times this refers to the randomly sampled variables used in the algorithm.

---

> ### Author Response · Authors · 2020-11-24
> **Response to AR3**
>
> We thank the reviewer for the feedback. In the new revision, we significantly polished the writing and believe that the current paper is easier to follow. We also added a visualization of our approach (see Figure 1).

---

### Official Review · AnonReviewer2 · 2020-10-31
**Interesting Idea**

**Rating:** 7
**Confidence:** 3

**Review:**

[Summary]
This paper proposes a new approach to learn disentangled representations from Generative models that were trained without a loss that explicitly enforces disengagement. The motivation behind this work is to address the computational time cost regarding hyperparameter tuning that is typically required for VAEs trained with a disentangled loss. The proposed approach can be summarized in 3 steps: (1) Training a GAN (in an unsupervised way) (2) Find  ‘k’ new basis vectors in ‘W’ space (3) Train a new encoder on the synthetic dataset {(z,G(z))} with loss that encourages the encodings to correspond to the subspace spanned by the basis vectors. Furthermore, the authors propose a new approach for identifying meaningful basis vectors which the authors claim that it is the generalized version of ClosedFrom. For evaluation, the authors then perform a set of experiments involving disentanglement, abstract reasoning, fairness on a variety of datasets such as 3Dshapes, MPI, and Cars3D.

[Clarity]
Overall, I think the paper reads well. I would however encourage the authors to add a background section that maybe explains a bit more regarding disentanglement, controllable generation, and in particular the style gan generator. The related work section touches on some of these, but not enough in my opinion.
As an example, I am personally not familiar with the GAN literature including the architecture of StyleGan decoder so I had to go and read [1] to see what is the difference between Z-space and W-space and what is the mapping.

Clarification on notation: In the related work section, the vector arithmetic is written as “z’ = z + a * n”. In the methodology, it is written as “w’ = w + a * n”. Is this just a change in notation, or ‘w’ here corresponds to the W-space? (If this is the case, why ‘w’ is later referred to as sample noise vectors. Also this doesn’t match the methodology Section in [2]).

[Main Comment]
Overall, I find the idea behind this paper quite interesting. While it is relatively simple and straightforward to implement, the idea of learning a disentangled representation from an entangled latent  is novel and makes sense.

Regarding the experiments section,  I applaud the authors for doing the fairness and abstract reasoning experiments in addition to reporting standard disentanglement metrics. I would however encourage the authors to directly compare with disentangled VAE baselines. While the caption Fig 1 claims that the results are better, the VAE results are not reported here. (I had to double check the results in [3] to make sure the results are indeed better).

One key weakness of their approach is that it relies on training the encoder based on ‘generated’ samples. This could mean that the proposed approach might not be very effective for models where the latent space might be reasonably disentangled, but the generated samples don’t look good (maybe blurry), given that the encoders job will presumnaly get harder as sample quality gets worse. However, as the authors state, this approach certainly can be applied for existing GANs that are known to generate high fidelity images.


[Questions]
- As far as I can see, this approach is not specific to GANs and can be applied to the latent space of VAEs as well. I’m wondering about the results from applying this approach to a trained VAE with a style gan decoder. Could the authors explain the focus on GANs?
- Could the authors elaborate on the proposed approach with the impossibility result in [1]. While the results suggest that we indeed were successful in learning a disentangled representation, it is not clear to me what inductive bias (or supervision) are we imposing here that encourages disentanglement. Is it simply the empirical evidence regarding controllable generation on GANs?
- It is unclear to me how the authors computed singular vectors at the deeper convolutional layers. Computing the Jacobian does not sound like an easy computation, and heavily depending on the base point in this case. Could the authors elaborate on this?

[References]
[1] Karras, Tero, Samuli Laine, and Timo Aila. "A style-based generator architecture for generative adversarial networks." Proceedings of the IEEE conference on computer vision and pattern recognition. 2019.
[3] Shen, Yujun, and Bolei Zhou. "Closed-Form Factorization of Latent Semantics in GANs." arXiv preprint arXiv:2007.06600 (2020).
[3] Locatello, Francesco, et al. "Challenging common assumptions in the unsupervised learning of disentangled representations." international conference on machine learning. 2019.


-------------
UPDATE
-------------

I thank the authors for their response. The revised version certainly looks better. I'm still happy to recommend this paper for acceptance.

---

> ### Author Response · Authors · 2020-11-24
> **Response to AR2**
>
> We thank the reviewer for the review and constructive comments. Here we address each of the concerns:
>
> [Could the authors explain the focus on GANs?] The main reason is that GANs achieve higher generation performance compared to VAEs, especially when it comes to small visual details. These details can correspond to essential factors of variation, crucial for proper disentanglement. For instance, Figure 1h in [A] demonstrates that VAE samples often do not capture the form of an object held by the robotic arm in the MPI3D dataset, which is one of the ground truth generative factors.
>
> [Elaborate on the proposed approach with the impossibility result in Locatello et al.] A strong inductive bias comes from the StyleGAN architecture, which is known to be an excellent model of the real image manifold (over 1300 citations since CVPR’2019). Another source of inductive bias comes from the success of controllable generation, as mentioned by the reviewer.
>
> [Computing singular vectors at the deeper convolutional layers] To compute singular vectors of the Jacobian, we use the iterative method (in fact, the one available in scipy), which requires only two operations: efficient multiplication of the Jacobian by a vector and the same operation for the Jacobian transposed. These operations can be very efficiently implemented (without forming the full Jacobian matrix) using automatic differentiation via the so-called Pearlmutter’s trick [B, C]. In practice, it takes only a few seconds to obtain the top 10 singular vectors, even for deeper layers of GANs we studied.
>
> [Direct comparison with disentangled VAE baselines] We agree entirely with this point and have added the direct comparison in Section 4 in the revised version.
> For added plots, we used the pretrained models for Cars3D provided in https://github.com/google-research/disentanglement_lib.
> We asked the authors (Locatello et al., ICML’19) for exact numbers for other datasets, but they could not provide them. Unfortunately, we cannot re-train thousands of models during the discussion phase, thus resort to a cross-paper comparison for 3DShapes.
>
> [More details on GANs and controllable generation.] Thanks. We have extended the “Related work” section and include the necessary details to make the submission fully self-contained.
>
>
>
> [A] [A Sober Look at the Unsupervised Learning of Disentangled Representations and their Evaluation](https://www.jmlr.org/papers/v21/19-976.html)
>
> [B] http://www.bcl.hamilton.ie/~barak/papers/nc-hessian.pdf
>
> [C] https://j-towns.github.io/2017/06/12/A-new-trick.html

---

### Author Response · Authors · 2020-11-24
**General comment**

We thank the reviewers for their time and useful suggestions. We have uploaded a new revision of our paper with changes described in the individual answers below. For this revision, we have trained a new set of StyleGAN 2 models for 64x64 datasets; specifically, we have now trained eight models for each, for double the number of iterations compared to our previous setup. This significantly improved the stability and quality of our method. Additionally, we have found a mistake in the ordering of labels provided for the $\texttt{Isaac3D}$ dataset. After fixing this bug, our scores significantly improved on this dataset as well. We have also polished the writing, focusing on clearer notations, a better explanation of our method, and making the paper more self-contained.

---

### Decision · Program_Chairs · 2021-01-07
**Final Decision**

**Decision:**

Reject

**Comment:**

This paper evaluates the extent to which disentangled representations can be recovered from pre-trained GANs with style-based generators by finding an orthogonal basis in the space of style vectors, and then training an encoder to map images to coordinates in the resulting latent space. To construct the orthogonal basis, the authors consider 3 recently proposed methods for controllable generation, along with a newly developed generalization of one of these methods. The authors evaluate metrics for disentanglement for 4 datasets, consider an abstract visual reasoning task, and compute unfairness scores.

Reviewers expressed diverging opinions on this paper. R2 is in support of acceptance,  R3 finds the paper borderline but is leaning towards acceptance, whereas R4 is critical. R2 and R4 engaged in a relatively detailed discussion, but maintained their scores.

Having read the paper, the metareviewer feels this submission indeed has strengths and weaknesses. On the one hand, the main results are notable; it is worth reporting that disentangled representations can be recovered from pretrained GANs is a relatively straightforward manner. In this context, the metareviewer feels that some comments by R4 are more critical than is warranted. The authors do not necessarily have to show that GAN-based methods uniformly improve upon VAE-based methods, either in terms of disentanglement metrics or in terms of sensitivity to hyperparameters. The main claim in this submission is that GAN-based methods are mostly comparable to VAE-based methods, and this claim is both sufficiently notable and sufficiently supported by experimental results.

At the same time, this submission is not without flaws.  The writing is on the rough side, and as R4 notes the authors have removed all white space between paragraphs. The metareviewer also feels it is not satisfactory to show a box plot for GAN-based methods in Figure 2 and ask the reader to compare these plots to the violin plots for VAE-based methods in the Locatello paper. The authors need to find a way to make a more direct comparison here. R4's comments about the comparison in the abstract-reasoning setting are also well-taken –– here the baseline employs standard (entangled) models, so it is unclear what conclusions we should draw from this experiment. Similarly the unfairness results once again appeal to an indirect comparison to results in the  Locatello paper on this topic.

On balance, the metareviewer is inclined to say that this submission, in its current form, falls just below the threshold for acceptance. These results are clearly of note to the community and worth reporting, but the presentation has enough flaws that another round of reviews is warranted based on a revised manuscript. The metareviewer hopes to see this paper appear a conference in the (near) future.